# Investigations of Olive Oil Industry By-Products Extracts with Potential Skin Benefits in Topical Formulations

**DOI:** 10.3390/pharmaceutics13040465

**Published:** 2021-03-30

**Authors:** Andreia Nunes, Lídia Gonçalves, Joana Marto, Ana Margarida Martins, Alexandra N. Silva, Pedro Pinto, Marta Martins, Carmo Fraga, Helena Margarida Ribeiro

**Affiliations:** 1Research Institute for Medicine (iMed.ULisboa), Faculty of Pharmacy, Universidade de Lisboa, 1649-003 Lisbon, Portugal; andreia.a.nunes@campus.ul.pt (A.N.); lgoncalves@ff.ulisboa.pt (L.G.); jmmarto@ff.ulisboa.pt (J.M.); amartins@farm-id.pt (A.M.M.); pfcpinto@ff.ulisboa.pt (P.P.); 2ADEIM, Laboratório de Controlo Microbiológico, 1649-003 Lisbon, Portugal; ansilva@ff.ul.pt; 3PhDtrials, Avenida Maria Helena Vieira da Silva, n° 24 A, 1750-182 Lisboa, Portugal; 4Marine and Environmental Sciences Centre (MARE), NOVA School of Science and Technology (FCT NOVA), Campus da Caparica, 2829-516 Caparica, Portugal; marta.martins@fct.unl.pt; 5Sovena Portugal—Consumer Goods, S.A., Rua Dr. António Borges n°2, Edifício Arquiparque 2-3° Andar, 1495-131 Algés, Portugal; carmo.fraga@sovena.pt

**Keywords:** olive oil industry by-products, extracts, creams, cytotoxicity, ecotoxicity, antioxidant

## Abstract

The by-products of olive oil industry are a major ecological issue due to their phenolic content, highly toxic organic load, and low pH. However, they can be recovered and reused, since their components have antioxidant, anti-inflammatory, and photoprotector properties. In this work, oil-in-water creams containing three different olive oil industry by-products extracts were produced without the use of organic solvents. First, the extracts were thoroughly characterized in vitro for cytotoxicity, inhibition of skin enzymes, and antioxidant and photoprotection capacities. Safety studies were then performed, including ocular and skin irritation tests, ecotoxicity evaluation, and in vivo Human Repeat Insult Patch Test. The results obtained in this initial characterization supported the incorporation of the extracts in the cream formulations. After preparation, the creams were characterized for their organoleptic, physicochemical, droplet size and rheological properties, and microbial contamination. The results showed that all formulations were semi-solid creams, with stable pH, compatible with the skin, without microbial contamination, and with the expected droplet size range. The rheological analysis showed shear-thinning behavior with yield stress, with the viscosity decreasing with increasing shear rate. The oscillatory results suggest that the creams have a strong network structure, being easily rubbed into the skin. Finally, compatibility, acceptability and antioxidant efficacy were evaluated in vivo, in human volunteers. No adverse reactions were observed after application of the formulations on skin and the cream with the highest concentrations of phenolic compounds showed the highest antioxidant efficiency. In conclusion, the results suggest that olive oil industry by-products extracts have valuable properties that favor their re-use in the cosmetic industry. The example presented here showed their successful incorporation into creams and their impact in these formulations’ appearance, pH, and rheological performance, as well as their in vivo compatibility with skin and antioxidant efficiency.

## 1. Introduction

The olive tree, also known as *Olea europaea*, is a polymorphous, medium-size tree, considered one of the most important fruit trees in the Mediterranean countries. Being native to Asia Minor and Syria, *Olea europaea*, has been cultivated in the entire Mediterranean area to produce olive oil and table olives, two of the most representative components of this diet and broadly consumed throughout the world [1,2,3].

Olive farming and the olive oil industry have grown over the past decades due to their important economic, environmental, and social impact in Mediterranean countries, where approximately 98% of the world olive oil is produced. Spain, Italy, Greece, and Portugal are considered the most representative olive oil producers in the world [4,5,6].

Despite the known economic, health and nutritional benefits of olive oil, this industry is associated with environmental issues, due to the large amounts of by-products generated during the olive oil production process. Among all the produced wastes, the olive pomace, olive leaves and olive mill waste waters (OMWW) are the main ecological concerns of this industry. These olive oil industry by-products, especially OMWW, are considered harmful for the environment mainly due to their phenolic content, highly toxic organic load, and low pH [4,5,7,8,9].

The olive pomace consists of a semi-solid to semi-liquid residue with a high content of water (60%), phenolic compounds and low pH. Additionally, this by-product contains considerable amounts of cellulose (30%) and pectic polysaccharides (39%) [5,8,10]. Olive leaves also contain high amounts of phenolic compounds, very similar to those present in the fruit and olive oil, such as secoiridoids (oleuropein and ligstroside), flavonoids (apigenin and luteolin), and phenolic compounds (tyrosol and hydroxytyrosol (HTyr)) [11,12]. The OMWW is a mildly acidic liquid with a red-to-black color, high conductivity, and mainly composed of water (83–92%). In addition, OMWW contains phenolic compounds (0.5–2.4%), mainly oleuropein and HTyr, but also secoiridoids derivatives, flavonoids, phenolic acids, and lignans [5,8,13].

To overcome this ecological issue, several applications for olive oil industry by-products have already been described. Due to their several bioactivities and health-promoting properties, these by-products are of a great interest for the pharmaceutical, cosmetic, and food industries. The recovery and reuse of these by-products and their bioactive compounds for industrial applications are a main feature of circular economy, where by-products are not considered waste, but a source of valuable compounds to be reused [2,4].

The olive oil industry by-products as well as their bioactive phenolic compounds have demonstrated antioxidant [9,14,15,16,17,18,19,20], anti-inflammatory [12,21,22,23,24,25,26,27,28,29], and photoprotector [15,26,30,31,32,33,34,35] activities. Additionally, their application has also been studied in the cosmetic and dermatology research field, for the treatment of skin diseases using different topical delivery systems [14,36,37,38,39,40]. Emulsions are one example of these delivery systems, which are usually composed by emollients, thickeners, humectants, preservatives, and water, among other ingredients. Emulsions are considered thermodynamically unstable and heterogenic systems, but are the most used vehicles in cosmetic industry, due to their efficacy, namely, hydration and emollience. Creams are an example of semi-solid emulsions. The incorporation of natural ingredients in creams, a tendency that has been increasing, may have an impact in their structure, leading to changes in their appearance and their physicochemical and rheological properties [41,42]. Therefore, the main aim of the present work was to evaluate the impact of incorporating three different olive oil industry by-products extracts (OIBPE) in skin care oil-in-water (O/W) creams, through their in vitro and in vivo characterization, and comparison with a control cream, not containing extracts.

## 2. Materials and Methods

### 2.1. Materials

#### 2.1.1. Olive Oil Industry By-Products Extracts

The OIBPE were a gift from Sovena Group (Algés, Portugal). In previous studies, the extracts exhibited health-promoting properties, possibly due to their antioxidant, anti-inflammatory, photoprotector, and antimicrobial activities. Since the olive harvesting includes a mixture of olives with branches and leaves that are used directly without any further processing and leaves are reintegrated in olive oil process by their deposition on soil to increase its content in organic matter and return nutrients then they are considered by-products. The OIBPE were obtained from leaves, without using organic solvents, either at high temperature (extracts 1 and 3) or at room temperature (extract 2). The extracts provided by our supplier (Sovena) were produced through an innovative process that are not shown due to confidentiality reasons. The main compounds detected in these extracts were determined by HPLC and are shown in Table 1.

#### 2.1.2. O/W Cream Formulation

In this work the extracts described in Table 1 were incorporated in O/W creams, with glycerin (Quimidroga, Lisboa, Portugal) used as a humectant, and 2-phenoxyethanol (Biopol PB 6, DS Produtos Químicos, Domingos de Rana, Portugal) as preservative. Additional ingredients were hydrogenated lecithin (and) C12–16 alcohols (and) palmitic acid (Biophilic™ H MB, IFF Lucas Meyer Cosmetics, Quebec City, QC, Canada), caprylic/capric triglycerides (Tegosoft^®^ CT, Evonik, Essen, Germany), Undecane (and) Tridecane (Cetiol^®^ Ultimate, BASF, Rhein, Germany), isopropyl myristate (Tegosoft^®^ M, Evonik, Essen, Germany), and purified water. Purified water was obtained by inverse osmosis (Millipore, Elix^®^ 3, Merck, Darmstadt, Germany).

### 2.2. Methods

#### 2.2.1. In Vitro Efficacy and Safety Analysis

##### In Vitro Cytotoxicity

The cytotoxicity of the OIBPE was evaluated on the human keratinocyte cell line HaCaT (Cell line Service GmbH, Eppelheim, Germany) and mouse fibroblasts cell line L929 (ATCC^®^ CCL-1™), using the endpoint MTT (3- (4,5-dimethyl-2-thiazolyl) -2,5-diphenyl-2H-tetrazolium bromide) reduction method [43,44]. The cells were seeded at 2 × 10^5^ cells/well in sterile 96-well culture plates (Greiner, Frickenhausen, Germany), in RPMI 1640 culture medium (Life Technologies, Warrington, UK), supplemented with 10% fetal bovine serum, 100 units of penicillin G (sodium salt) and 100 μg of streptomycin sulfate, and 2 mM L-glutamine (Life Technologies, Warrington, UK), at 37 °C and 5% CO_2_. The cells were incubated with different concentrations of the extracts (0.1–0.0015 mg/mL) for 24 h in the same conditions. Cells incubated with culture medium were used as negative control, while incubation with 1 mg/mL sodium dodecyl sulfate (SDS) was used as positive control. After the exposure time, the medium was replaced with medium containing 0.5 mg/mL MTT, and the cultures were incubated in the same conditions for 3 additional hours. The medium was removed, and the intracellular formazan crystals were solubilized and extracted with dimethyl sulfoxide (DMSO). After 15 min at room temperature, the absorbance was measured at 570 nm in a microplate reader (FLUOstar Omega; BMG Labtech GmbH, Ortenber, Germany). The cell viability (%) compared to control cells was calculated by (Absorbance) sample/(Absorbance) control × 100, culture medium with the same amount of DMSO as the samples used as control. The (IC_50_) inhibitory concentration that reduce cell viability to 50% was determined by non-linear regression analysis using the GraphPad program (GraphPad PRISM 5.01 software, La Jolla, CA, USA) using the data of log of extracts concentrations versus the percentage of cell viability.

##### Human Neutrophil Elastase

Fluorometric inhibition assays for Human Neutrophil Elastase (HNE, Merck, Darmstadt, Germany) were conducted as previously described [43], in 0.1 M HEPES pH 7.5 (assay buffer) at 25 °C, in a total volume of 200 μL (containing 20 μL of 0.17 μM HNE in assay buffer, 155 μL of buffer and 5 μL of each extract sample at a final concentration of 0.025 mg/mL).

The reaction was initiated by adding 20 μL of 200 μM fluorogenic substrate (MeO-Suc-Ala-Ala-Pro-Val-AMC, Merck, Darmstadt, Germany), and the activity was monitored for 30 min at 25 °C on a microplate reader (FLUOstar Omega; BMG Labtech GmbH, Ortenber, Germany) (excitation wavelength 380 nm, emission wavelength 460 nm). Controls were performed as follows: (1) enzyme, (2) substrate, (3) enzyme with DMSO, and (4) positive control with Sivelestat sodium salt hydrate (Sigma Aldrich, Gillingham, UK). Assays were performed in six independent experiments (*n* = 6) and presented as log of inhibitor concentrations versus the percentage of activity. IC_50_ values were determined by non-linear regression analysis in GraphPad PRISM^®^ 5 software.

##### Collagenase

The collagenase activity was determined using the synthetic peptide *N*-(3-(2-furyl)acryloyl)-Leu-Gly-Pro-Ala (FALGPA) (Sigma-Aldrich, Madrid, Spain) to simulate collagen, as described by Roda et al. [45]. The inhibition studies were performed as follows: 10 µL of OIBPE were added (to a final concentration of 0.025 mg/mL) to a solution of 10 µL of collagenase (0.35 U/mL) (Sigma-Aldrich, Spain) and 80 µL of assay buffer (50 mM TES, 0.36 mM CaCl2, pH 7.4); the reaction was then started with the addition of 40 µL of FALGPA and 60 µL of buffer. The absorbance was measured at 345 nm for 5–15 min in a microplate reader (FLUOstar Omega; BMGLabtech GmbH, Ortenber, Germany).

##### Hyaluronidase

The OIBPE capacity to inhibit hyaluronidase activity was evaluated according to the method described by Sigma Aldrich [46]. The assay was performed in a reaction mixture containing 0.015% (*w*/*v*) hyaluronic acid, 150 mM sodium phosphate and 2–5 units of hyaluronidase (Sigma-Aldrich, Madrid, Spain), to which 10 μL of OIBPE were added. After incubating the samples at 37 °C for 45 min, 0.1 mL of each sample was transferred to a 96-well plate containing 0.1 mL of an acid albumin solution. After 10 min at room temperature, the absorbance was measured at 600 nm in a microplate reader (FLUOstar Omega; BMGLabtech GmbH, Ortenber, Germany).

##### Tyrosinase

The inhibition of mushroom tyrosinase activity by the OIBPE was determined according to a previously described method [47], with some modifications. The reaction mixture, added to a 96-well microplate, was composed of 151 μL of 0.1 M sodium phosphate buffer (pH 6.5), 5 μL of OIBPE (Final concentration of 0.025 mg/mL) dissolved in buffer, 8 μL of mushroom tyrosinase (Sigma-Adrich, Madrid, Spain) (2500 units/mL), and 36 μL of L-tyrosine (5 mM). The reaction mixture was incubated at 37 °C in a microplate reader (FLUOstar Omega; BMGLabtech GmbH, Ortenber, Germany) and the absorbance was measured at 280 nm every minute for 30 min. The activity of the tested samples was determined by the initial rate of dopachrome formation of the reaction mixture and compared with the activity of the buffer solution.

##### Antioxidant Capacity

The intracellular production of reactive oxygen species (ROS) within cells was evaluated with a fluorometric technique using 2,7’ dichlorodihydrofluorescein diacetate (H_2_-DCFDA, Life Technologies, Warrington, UK). The H_2_-DCFDA is a stable, non-fluorescent molecule that is hydrolyzed by intracellular esterases to non-fluorescent 2-7′-dichlorodihydrofluorescein (H_2_DCF), which is rapidly oxidized in the presence of H_2_O_2_, hydroxyl radicals, and diverse peroxides [48] to a highly fluorescent compound (DCF) [47,49].

HaCaT sub-confluent cells seeded in 96-well plates were incubated for 30 min with 20 μM of H_2_-DCFDA at 37 °C. The culture medium was removed, and fresh medium was added to the cells before being exposed to two different concentrations (0.1 and 0.05 mg/mL) of OIBPE or 1 mg/mL ascorbic acid (positive control) for 1 h. The oxidative stress was induced in cells using a 500 μM hydrogen peroxide (H_2_O_2_) solution or by exposure to UVB light (emission wavelength 312 nm) for 15 min. After exposure, ROS levels were determined by fluorescence (excitation wavelength 485 nm, emission wavelength 520 nm), in a microplate reader (FLUOstar BMGLabtech, Ortenber, Germany). Data from 9 replicates (*n* = 9) are reported as the percentage of ROS reduction determined as: 100-(fluorescence of exposed cells/fluorescence of unexposed control from the same experiment) × 100.

##### Photoprotection Studies

To evaluate the photoprotector activity of the OIBPE, a spectrophotometry method was used to calculate the sun protection factor (SPF), as described by Mansur et al. [50], as a screening test. This methodology measures the absorption characteristics of the sunscreens’ agents based on spectrophotometric analysis of dilute solutions. Moreover, the method uses the erythema effect spectrum (“erythema formation”) and solar intensity spectrum constant values from 290 nm to 320 nm, being possible to correctly accurate de in vitro SPF value of the extracts.

The OIBPE were diluted (1:40) in distilled water in a 96-well plate (*n* = 3). The absorption spectrum was measured from 290 nm to 320 nm every 5 nm, with 3 measurements per point. Distilled water was used as control. The spectra were obtained using a fluorescence microplate reader (FLUOstar BMGLabtech, Ortenber, Germany).

The results were analyzed in Microsoft Excel, according to the Equation (1) described by Mansur et al. [50]:(1)$$SPF=FC\;\times \;\sum_{290}^{320}EE\;\left(\lambda \right)\times I\;\left(\lambda \right)\times Abs\;\left(\lambda \right),$$
where *FC* corresponds to the correlation factor (nominal value 10) determined according to two known SPF sunscreens; *EE* (λ) represents the effect of “erythema formation” at = wavelength (λ); *I* (λ) correspond to the intensity of the sun at the wavelength (λ); and *Abs* (λ) represents the spectrophotometric reading of the absorbance of the solution in the filter at the wavelength (λ) [51,52,53].

##### Antimicrobial Activity

The antimicrobial studies were performed to evaluate the minimum inhibitory concentration of three extracts against for five ATCC strains, namely, *Staphylococcus aureus* ATCC 6538, *Pseudomonas aeruginosa* ATCC 9027, *Bacillus subtilis* ATCC 6633, *Escherichia coli* ATCC 8739, and *Candida albicans* ATCC 10231.

The assay was performed in 96-well sterile culture plates, containing Sabouraud Dextrose Broth (BIOKAR Diagnostics, Allonne, France) (for *C. albicans*) or Mueller Hinton Broth (BIOKAR Diagnostics) (for all other strains) as culture media. The microbial suspensions were prepared in sterile APT broth (BIOKAR Diagnostics) for bacteria and mold, to a concentration of approximately 1.5 × 10^8^ CFU/mL. After incubation with the OIBPE at (32.5 ± 2.5) °C for 48 h, the absorbance was determined at 600 nm using a fluorescence microplate reader (Varioskan ™ LUX, Thermo Scientific ™, Waltham, MA, USA).

##### Safety Studies: Ocular and Skin Irritation

The eye irritation test was performed according to the method based on the Human Corneal Epithelium (HCE) model (SkinEthic ™ HCE—Eye Irritation Test Liquid, Lyon, France [54]. The model is characterized by the presence of immortalized corneal epithelial cells—HCE, 0.5 cm^2^ of reconstructed epithelium from transformed keratinocytes of the human cornea and cells grown in an inert polycarbonate matrix in a defined medium for 5 days. Two independent experiments (*n* = 2) were performed with the OIBPE, using phosphate-buffered saline (PBS) as a negative control and methyl acetate as a positive control. The results obtained are presented as a percentage of cell viability following the acceptance criteria defined by the method guideline No. 492 of the OECD [54].

The SkinEthic ™ HCE method—Eye Irritation Test Liquid is not intended to differentiate substances between category 1 (severe eye damage) and category 2 (eye irritation); thus, according to the method described above, it was possible to classify the irritation potential of the OIBPE into two distinct groups: “No category” and “Category 1 or Category 2”. A substance is considered as “No category” if the relative average viability of the tissue exposed to it is higher than 60%.

The skin irritation test was performed according to the method based on the Reconstructed Human Epidermis (RHE) model (EpiSkin ™ Skin Irritation Test—Episkin ™: Small Model, Lyon, France) [55]. The model is characterized by the presence of 0.38 cm^2^ of epidermis reconstructed with normal human keratinocytes and cells grown in a collagen matrix for 13 days. The test was performed with the OIBPE in three independent experiments (*n* = 3), using PBS as a negative control and SDS as a positive control. The results obtained are presented as a percentage of cell viability following the acceptance criterion defined by the method guideline No. 439 of the OECD [55].

Additionally, the irritation potential of the substance is determined according to UN GHS and EU CLP as “Category 2: Irritant” or as “No category” [56]. A substance is considered as irritant if the relative average viability of the tissue exposed to the sample is reduced to less than 50% of the average viability of the negative control.

##### Ecotoxicological Evaluation

The evaluation of the extracts ecotoxicity was carried out according to [57], and conducted according to the OECD Guidelines for chemical safety adopted by FCT NOVA. The method, called Microtox^®^ Acute Toxicity Test, uses a marine bacterium *Aliivibrio fischeri* to detect the toxicity of a mixture of compounds, measured as inhibition of the bacteria bioluminescence. The analysis was carried out using the Microtox 500 analyzer (ModernWater). *A. fischeri* was exposed to crude extracts and their dilutions (up to 1 × 10^6^ times) and the bacterial luminescence was measured at 5, 15, and 25 minutes. The extracts without bacteria were also displayed to check the possible interference of the matrix in the reading of bioluminescence (negative control). The results were expressed as the concentration that causes the loss of 50% of the bacterial luminescence (EC_50_) using the MicrotoxOmni software (version 4.3, 1995, Los Angeles, CA, USA).

#### 2.2.2. In Vivo Safety Study: HRIPT

To evaluate the safety of the extracts, when applied under normal conditions of use, the Human Repeat Insult Patch Test (HRIPT) was performed for all extracts.

For this study, 51 healthy volunteers were chosen, who were informed about the procedure and signed the informed consent. The specific non-inclusion criteria of the simple patch test were applied. The Ethical approval code are 15471120.T; 15471120.U and 15471120.V done in 13/01/2021. The study was executed according to the Marzully and Maibach HRIPT protocol, as previously described by Couto et al. [58]. The study was performed for six weeks divided in three different phases: (1) the induction phase that was carried on for three weeks, (2) the two weeks rest phase, and (3) the final week of study. In the first phase, 20 μL of each extract aqueous solution (5%, *w*/*v*) were applied and protected by an occlusive adhesive (Finn Chamber standard, Smartpractice), in the back of each volunteer. The product was kept in contact with the skin for 48 h or 72 h, before being removed and the skin reaction analyzed. The extract application was repeated nine times in the first phase, followed by two weeks during which no product was applied. In the third phase, the product was applied in the same place and in a complementary place where no previous application had occurred. The extract was removed after 48 h and the skin reactions at 48 h, 72 h, and 96 h after application, were evaluated.

#### 2.2.3. Development and Characterization of Formulations Containing Olive Oil Industrial By-Products Extracts

##### Preparation of the Formulations

Three O/W creams were prepared using three different OIBPE, as well as a control formulation consisting of a cream without any added extract. The 5% OIBPE concentration was used as the worst possible scenario regarding the impact of these extracts on the structure of the formulations.

The formulations were prepared using a hot emulsification method (80 °C), where the oily and aqueous phases were melted separately. Thereafter, the oily phase was added to the aqueous phase at 80 °C and mixed with a vertical mixer (IKA^®^ T25 Digital—ULTRA-TURRAX^®^, Staufen, Germany) at 8000 rpm for 5 min, followed by stirring using a vertical stirrer (IKA^®^ EUROSTAR 60, digital, Staufen, Germany) at 400 rpm, until reaching room temperature (27–30 °C). The composition of the formulations is shown in Table 2.

Considering previous studies accomplished by our group (data not shown) the creams with the compositions shown in Table 2 were selected to pursuit these studies due to their physicochemical and sensorial properties, and higher stability.

##### Physicochemical and Microbiological Studies

All formulations’ organoleptic characteristics, such as appearance, color, odor, and pH, were evaluated. The pH values were determined at room temperature using a pH-meter (Mettler Toledo, Columbus, OH, USA) in the production day (T_0_) and 6 days after production (T_6_).

To evaluate the microbiological contamination of all formulations, microbiological control tests were carried out using a spread-plating method, always in sterile conditions. In this assay, Trypticase Soy Agar (TSA) (BIOKAR Diagnostics) and Sabouraud Dextrose Agar (SDA) (BIOKAR Diagnostics) were used as media. After autoclaving, the media were distributed by Petri plates with 90 cm in diameter (VWR). The formulations were diluted 1:10 and 1:1000 in sterile Buffered Peptone Water (BIOKAR Diagnostics) pH 7.0. The TSA and SDA media plates were inoculated with 100 μL of each sample dilution (n = 1) and were incubated at 30 °C for the SDA medium and at 37 °C for the TSA media, for 3–5 days. The number of colony-forming units (CFU) that developed during the incubation period (3–5 days) was evaluated and recorded. The count of total microorganisms corresponds to the average of CFU present in the TSA and SDA plates multiplied by the dilution factor.

##### Droplet Size Analysis

The formulations were analyzed for droplet size using an optical microscope with polarized light (Nikon eclipse C*i*, Nikon, Japan) incorporated with a camera (Sony Exmor CMOS Sensor, MicroCopiaDigital, Münster, Germany). For better visualization, the emulsion samples, previously diluted in water, were directly applied on the blade. The droplet size distribution was studied using a Malvern Mastersizer 2000 with a Hydro 2000S module (Malvern, Kassel, Germany) 6 days after production. Additionally, the parameters used in this method were previously optimized. Briefly, an obscuration between 10 and 20% and an agitation of 1750 rpm were used as experimental parameters. The results obtained were analyzed using Microsoft Excel software (version 16.46, Redmond, WA, USA).

##### Rheological Analysis

The rheological characteristics of the formulations were examined 9 days after the production, at high shear rates using continuous shear techniques and in the viscoelastic region using oscillation techniques. These experiments were performed with a controlled stress Kinexus Lab^+^ Rheometer (Malvern Instruments, Worcestershire, UK) using cone-plate geometry (truncated cone angle 4° and radius 40 mm). All the studies were performed at room temperature.

The shear rate method (table of shear rates) was performed using a destructive measurement, where the shear stress of each emulsion was obtained by increasing the shear rate from 0.1 s^−1^ to 100 s^−1^. Similarly, a flow curve method, where the shear rate increased and decreased from 0.1 s^−1^ to 100 s^−1^ for 10 min, was accomplished. Additionally, the viscosity behavior of the formulations was characterized by fitting rheological models (Power-law and Herschel–Bulkley models) using the Kinexus Lab^+^ Rheometer program rSpace for Kinexus (Version 1.76.2398.0, 2019, Worcestershire, UK). To choose the most appropriate model the correlation coefficient was considered.

Regarding the oscillatory method, first an amplitude sweep test was performed where the shear strain ranged between 0.001 and 100% and the frequency was set at 1 Hz. Then, a frequency sweep test was performed with a shear strain of 0.1% and a frequency range between 0.1 and 10 Hz. All the experiments were performed using 10 samples per decade.

##### In Vivo Compatibility, Acceptability, and Antioxidant Efficacy

This study aimed to check the compatibility and acceptability of the formulations, and to assess their antioxidant efficacy.

The cutaneous acceptability and compatibility studies were supervised by a dermatologist, who evaluated the results after visual examination of the experimental area and the questioning of the subjects. The antioxidant efficacy of the topical formulations was evaluated by the in vivo assay, described by Couto et al. [58], which uses colorimetry to monitor changes in the chromophore capability of β-carotene applied to skin, after exposure to UVA. The skin color was determined using a tristimulus color analyzer that measures the reflected color. A Chromameter CR-400^^®^^ (Minolta, Tokyo, Japan) was used to detect any slight deviation in the xenon’s light spectral distribution. The system provides data for the luminance (L*), a* (red-green), and b* (blue-yellow) color distribution.

For the antioxidant efficacy assay, 10 healthy female volunteers aged 18–65 years were chosen. All volunteers were informed about the procedure and signed an informed consent form. The same specific criteria for non-inclusion of the test were applied. The study code is 15661220.A-D accomplished in 14 January 2021. This is a single-center, blinded controlled study in healthy subjects. In the beginning of the study, four areas in one forearm were treated with the products under investigation, while the other forearm, left untreated, was used as a positive control (solution of β-carotene). Subsequently, a solution of β-carotene was applied in both arms. A blank skin area was used as a negative control. The color was measured before and after 1 hour of exposure to UVA irradiation (1 J/cm^2^). β-carotene is a yellow chromophore molecule that when oxidized loses its chromophore capability and color, and this discoloration can be monitored by colorimetry (b* parameter).

#### 2.2.4. Statistical Analysis

Data distribution analysis was determined by the Shapiro–Wilk test. For normal data one way ANOVA was performed, followed by multiple comparisons using Tukey’s test. For other non-parametric analysis, the Kruskal–Wallis test was performed, followed by multiple comparisons using Dunn’s tests. The mean and standard deviation (SD) are presented. A *p* < 0.05 was considered statistically significant. All data analysis was done with GraphPad program (GraphPad PRISM 5.01 software, La Jolla, CA, USA).

For the in vivo experiments, the instrumental efficacy data were expressed in numbered data and submitted to a suitable statistical treatment. A comparative analysis performed by the Wilcoxon Signed Ranks Test for paired data (non-parametric) was used if the distribution was not normal, and a paired sample t-student test was used if the distribution was normal. In both cases the significance level adopted was 95%. All the calculations were performed using SPSS 23 (IBM). The subjective data of efficacy was submitted to a suitable statistical treatment Binomial test and Chi-square test.

## 3. Results and Discussion

### 3.1. In Vitro and In Vivo Efficacy and Safety Analysis

Olive oil industry by-product extracts present several health promoting properties, having been described as antioxidants, photoprotectors, anti-inflammatory, and antimicrobial agents in cosmetics or pharmaceutical products.

Therefore, in this work, several screening studies were performed for the OIBPE, such as in vitro cytotoxicity, enzymatic inhibition, antioxidant properties, and photoprotection activity, as follows.

#### 3.1.1. In Vitro Cytotoxicity Assay

The in vitro assays to determine the cytotoxicity of extracts 1, 2, and 3, were performed on HaCaT and fibroblasts L929 cells and the results are summarized in Table 3. Both assays aimed to study the possible cytotoxic effects of the extracts, and cytotoxicity was evaluated using a MTT reduction method.

The results of IC_50_, the concentration of extract needed to reduce the cell viability to 50%, show that exposure to different OIBPE have different effects on both cell lines. Extract 2 showed the lowest cytotoxicity for both HaCaT and L929, with IC_50_ higher than 0.1 mg/mL. Extracts 1 and 3 have the same effect in HaCaT cells, while extract 1 had a higher cytotoxic effect in the L929 cell line, with the lowest IC_50_ value, 0.016 mg/mL.

The results obtained can be related to the concentrations of total phenol compounds and of HTyr present in each extract (Table 1). Extract 2, which presents the lowest cytotoxicity for both cell lines, has the lowest phenolic concentration and the lowest concentration of HTyr (4488 mg GAE/L and 135 mg/L, respectively).

#### 3.1.2. Inhibition of Enzyme Activities

To study the use of OIBPE as potential anti-ageing and anti-inflammatory agents, the effects in elastase, collagenase, hyaluronidase, and tyrosinase activities were determined using fluorometric methods. The results of enzyme activity inhibition are shown in Table 4.

The results obtained show that all OIBPE present less enzymatic activity inhibition for collagenase, hyaluronidase, and tyrosinase, than for elastase. It is important to highlight that the OIBPE sample 2 and 3 were active, even at low concentrations, to inhibit enzymes involved in inflammatory and aging processes.

In this first approach, and in the assayed conditions, all the fractions presented some degree of anti-enzymatic activity, at least over one of the tested enzymes. All extracts were able to efficiently inhibit elastase activity without differences significantly different, with inhibition percentages varying between 94% and 100%. These results are considered promising in topical formulations since these extracts can protect skin cells from elastase, which is involved in the metabolism of elastic fibers, and thus it is related with the decrease in skin elasticity and consequent formation of wrinkles that occur during aging [30].

#### 3.1.3. Antioxidant Capacity

To study the OIBPE’s antioxidant capacity, i.e., their ability to reduce the formation of ROS, an in vitro antioxidant assay was performed. This fluorometric assay was performed using HaCaT cells seeded in 96-well plates, and the intracellular ROS formation was induced either by exposure to 500 μM H_2_O_2_ or to UVB light for 15 min.

The results obtained (Figure 1) show that all extracts were able to reduce ROS formation in HaCaT cells. In cells exposed to UV, the % of ROS reduction with extracts 1, 2, and 3 were not significantly different from each other but were significantly lower than the % of reduction obtained with ascorbic acid. In cells exposed to H_2_O_2_ the % of ROS reduction with extract 2 was significantly higher than the one observed for extract 1, but not for that obtained for extract 3. The % of reduction obtained with extracts 1 and 3 were significantly lower than that of ascorbic acid. However, in both oxidative stress conditions, all extracts were able to reduce the ROS formation by more than 80%, thus they all demonstrated high antioxidant activity in the HaCaT cell line exposed to oxidative stress. These results are in accordance with previously reported results [9,14,15,16,31,32], of studies that tested OIBPE as antioxidants in oxidative stress conditions induced by H_2_O_2_, UVB light, and other commonly used methods such as DPPH (2,2-diphenyl-1-picrylhydrazyl) and ABTS (2,2’-azino-bis(3-ethylbenzothiazoline-6-sulfonic acid). Nevertheless, it is important to consider that the concentration of phenolic compounds and HTyr in the extracts described in the literature and in the extracts studied herein is not the same; thus, differences in results are expected.

#### 3.1.4. Photoprotection Studies

The in vitro photoprotection assays were performed study the extracts’ photoprotection capacity and their possible use as photoprotector adjuvants. This assay was accomplished using a spectrophotometry method described by Mansur et al. [50], which allows to calculate the SPF. Extracts 1, 2, and 3 showed SPF values of 5.2 ± 0.3, 10.1 ± 0.2, and 7.1 ± 0.3, respectively, with the higher SPF value obtained for extract 2. No relationship can be observed between the SPF of the extracts and their concentrations of phenolic compounds (Table 1).

By-products of olive oil industry can be used in cosmetics as sun photoprotector boosters, as has already been reported in the literature. Galanakis et al. [15], for example, have reported that phenols from OMWW absorbed both UVB and UVA radiations and suggested their use as UV-protection boosters to increase the absorption of some synthetic UV filters, such as benzophenone-3, octocrylene and octyl methoxy cinnamate. Another study by the same author [38] also using phenolic compounds recovered from OMWW, reported results showing that the absorption and the in vitro SPF values of the UV filter solutions increased linearly as a function of phenol concentrations.

The results obtained in this work also support that the OIBPE may act as UV-protection boosters to increase the absorption of some synthetic UV filters, even if no linear relationship was observed with the phenol concentrations.

#### 3.1.5. Antimicrobial Activity

The minimum inhibitory concentration (MIC) assay was carried out to evaluate the antimicrobial capability of the OIBPE, against five ATCC microbial strains, which are considered the most relevant and representative strains of bacteria and pathogenic yeast. The results are shown in Table 5.

The results show that all OIBPE have antimicrobial activity against (at least some) microbial strains, with extract 3 being able to inhibit the growth of all the tested ATCC strains. *B. subtilis*, *P. aeruginosa* and *E. coli* were the most susceptible organisms to extract 1. *B. subtilis* is a Gram-positive bacterium which form endospores, while *P. aeruginosa* and *E. coli* are Gram-negative bacteria. On the other hand, only the Gram-positive *S. aureus* and the Gram-negative *P. aeruginosa* were susceptible to extract 2. Considering the only fungal strain tested, the pathogenic yeast *C. albicans*, only extract 3 was able to inhibit its growth, suggesting that extracts 1 and 2 might be antibacterial only. However, further studies with other fungal strains that may contaminate cosmetics, such as *Aspergillus niger*, *Cladosporium* spp. and *Penicillium* spp., would be necessary to support such a conclusion.

The phenolic content of the extracts can be related to their antimicrobial capability, since extract 3, with the highest concentrations of phenolic compounds (6127 mg GAE/L), also has the highest antimicrobial activity, inhibiting the growth of all tested strains, with MIC ranging between 1:2 and 1:8 dilutions.

In conclusion, the results obtained for the antimicrobial activity of the tested OIBPE suggest their possible use in the cosmetic and/or pharmaceutical industries as preservative boosters, either separately (e.g., extract 3) or combined.

#### 3.1.6. Safety Assays

##### Ocular and Skin Irritation

The SkinEthic™ and the EpiSkin™ assays are two well-known and valuable methods, which are regulated by OECD guidelines. Both methods are used worldwide to evaluate the potential irritation caused by a certain sample in eye and skin cells and classify them as “No category” or “Category 1 or 2”. The three OIBPE studied in this work were evaluated using these methods and the results are shown in Figure 2.

The assays were performed in accordance with the acceptance criteria of the validated method for both SkinEthic™ and EpiSkin™ small model. The preliminary test also showed that there was no interference between the tested samples and the MTT solution.

The results obtained for the SkinEthic™—Eye irritation test showed a mean tissue viability higher than 97% in the presence of the tested OIBPE, thus, no eye irritation could be detected in this in vitro assay. These results support the classification of all OIBPE as “No Category” (No. 1272/2008, EU, 2008), according to the guideline [54].

The results for the EpiSkin™—Skin irritation test showed a mean tissue viability higher than 65% for all the extracts. Moreover, the tested samples did not show skin irritation in this in vitro assay of reconstructed epidermis with normal human keratinocytes. Therefore, and because the mean tissue viability percentage is higher than 50%, the OIBPE are classified as “No Category” (UN, 2009; No. 1272/2008, EU, 2008) in accordance with the guideline [55].

##### Ecotoxicological Evaluation

The microtox toxicity test results are shown in Table 6. The crude extracts caused 100% toxicity to the *Aliivibrio fischeri* bacteria. Extracts 1 and 3 still caused 100% toxicity after being diluted 1:1000 (results not shown in the Table). The analyzed matrices (control) did not interfere in the luminescence readings.

In summary, the results show that the three extracts have different ecotoxicities, with extract 2 being the less ecotoxic, while extract 1 showed the highest ecotoxicity. Furthermore, acute toxicity did not depend on the exposure time.

##### In Vivo Safety

The in vivo safety assay was performed in order to confirm skin compatibility and the absence of delayed skin sensitizing potential of the extracts, after repeated application on human skin, under experimental conditions that involve over-exposure (patch).

In the experimental conditions adopted, the repeated applications of extracts 1, 2, and 3, in an occlusive patch, did not induce any irritation reactions. Moreover, the extracts presented good cutaneous compatibility. Furthermore, the repeated applications of all extracts did not induce any allergic reaction.

### 3.2. Formulations Containing Olive Oil Industry By-Products Extracts

Based on the results so far obtained, the OIBPE used in this work were considered very promising extracts for skin health. Therefore, they were incorporated in three oil in water (O/W) creams and their impact in cosmetic formulations was evaluated.

#### 3.2.1. Physicochemical and Microbiological Studies

The macroscopic examination of the emulsions containing the by-products extracts showed that all formulations presented a beige appearance and had the extracts’ characteristic smell, while the Control Cream showed a white color and had no smell. Additionally, all the formulations exhibited an opaque appearance with some consistency, being considered a semi-solid cream.

The results from pH determination showed that all creams presented a pH in the range 4.5–5.5, which was stable for six months. It is of the utmost importance that all formulations respect the skin natural pH, thereby all the formulations should present an acidic-neutral pH [59]. Thus, the values obtained are acceptable according to the established criteria, i.e., epidermal pH. Moreover, the formulations are considered stable and suitable for topical application since the pH values did not suffer any significant modifications over time.

Regarding the microbiological studies, no microorganism contamination was observed for any formulation.

#### 3.2.2. Droplet Size Analysis

The droplet size distribution characterization was studied to evaluate the influence of the incorporation of OIBPE in the droplet size of the cream formulations (Table 7).

Creams 1, 2, and 3 have d (0.5) values of 35.0 μm, 44.3 μm, and 43.6 μm, respectively, meaning that about 50% of the formulations’ droplets present the mentioned droplet size. The average droplet size for these formulations is higher than that of the Control Cream (*d* (0.5) = 16 μm), possibly due to the incorporation of the extracts in the formulations. Additionally, the droplet size results are in accordance with the results obtained in the literature for this type of formulations, which usually ranges from 0.1 to 50 μm [42,43,60,61].

Representative photomicrographs of the four creams are shown in Figure 3. In ordinary light, distorted oil droplets are visible in all the analyzed creams. The Control Cream contained smaller droplets than Cream 1, whereas droplet sizes were larger in the creams containing extracts 2 and 3. Polarized light microscopy is suitable for detection of lyotropic liquid crystals because liquid crystals show birefringence just like real crystals. Crystalline structures were not prominent in these creams meaning that lamellar structures were not present.

Comparing the results obtained with the composition of the OIBPE (Table 1), there seems to be no direct relationship between the concentration of phenolic compounds presented in the extracts and the droplet sizes. However, oleuropein is a molecule that can undergo hydrolysis with heating, favoring the appearance of various compounds such as HTyr, elenolic acid. This oleuropein possible degradation may be the cause of the breakdown of emulsions’ internal phase droplets. In fact, Stamatopoulos et al. [62] used an emulsifier (Tween 80) to inhibit a high degradation of oleuropein. Furthermore, the other minor components (data not shown) present in each extract may influence the formulations’ droplet size.

#### 3.2.3. Rheological Characterization

The rheological characterization was performed at high shear rates by continuous shear and oscillation experiments. Figure 4 shows the results of viscosity studies, in which the system response was measured as a function of shear rate.

The different creams presented different apparent viscosities at the same shear rate. Cream 2 showed the lowest apparent viscosity (28.25 Pa·s at 0.1 s^−1^), while Cream 1 had the highest apparent viscosity (59.76 Pa·s at the same shear rate value). Additionally, Cream 3 exhibited an apparent viscosity of 34.08 Pa·s at 0.1 s^−1^ shear rate. The comparison of these results with those of the Control Cream, which showed an apparent viscosity of 52.97 Pa·s at the same shear rate, suggest that the incorporation of the extracts influenced the internal structure of the emulsion. Nevertheless, the incorporation of extract 1 does not seem to significantly alter the formulation’s viscosity, since it shows a similar behavior to that of the control formulation. As discussed in the previous section, these results may be due to a possible degradation of oleuropein and the presence of minor compounds in the extracts. Furthermore, all formulations are considered as Non-Newtonian fluids, which is in accordance with previously obtained results [63,64,65]. Due to their behavior, the formulations are classified as shear thinning emulsions, which can be easily rubbed into the skin. In shear thinning emulsions, the viscosity decreases under a shear rate, mainly due to their molecular semi-flexible structure.

The results for the flow curve method (Figure 4) show the emulsions’ system response when the shear rate increase (up-curve), gradually breaking down the formulation’s structure, as well as the emulsions’ behavior when the shear rates decrease (down-curve), to allow the structure recovery. It is possible to observe the presence of a hysteresis loop indicating that all creams are thixotropic materials. This thixotropy, which is characteristic of more cohesive structures, is more pronounced in the Control Cream.

The formulations’ viscosity behavior was further characterized by fitting mathematical models, such as the Power-law and the Herschel–Bulkley models, to the rheological data (Table 8).

The best fitting was obtained with the Herschel–Bulkley model, with a correlation coefficient equal to or higher than 0.996 for all formulations. This model describes a material with a non-linear behavior identifying the power law index (*n*), which describes a material as shear thickening (*n* > 1) or as shear thinning (*n* < 1) [66]. In a set of renowned publications, Princen [67,68] studied the behavior of these hyper concentrated emulsions in flow. In particular, he demonstrated that, beyond the critical stress, the rheogram obeyed the Herschel–Bulkley equation. The results obtained herein show that the power law index varies between 0.410 and 0.559. Therefore, since the power law index is lower than 1 (*n* < 1), this model confirms that all cream formulations can be considered as shear thinning fluids.

Furthermore, the yield stress value is an important parameter of formulations since it is a predictor of the product stability. Higher yield stress values indicate that a formulation has higher stability and is less likely to experience phase separation, sedimentation, or aggregation [66,69,70]. The Control Cream formulation showed the highest yield stress value, namely, 1.459 Pa, meaning that this emulsion presents a consistent and a cohesive structure, with a possible higher stability than the formulations containing the extracts. It may be hypothesized that these results may be due to interactions between components of the incorporated extracts and the formulation, which might change their internal structure and, consequently, their stability.

The oscillatory study (Figure 5) represents the measured system response as a function of frequency at constant shear strain. Additionally, it provides information about two important components: the storage modulus or elastic component (*G*′) and the loss modulus or viscous component (*G*″).

The results show that in all formulations the storage modulus overcomes the loss modulus (*G*′ > *G*″). Additionally, neither elastic nor viscous component showed any dependency on frequency, being almost constant with the increase of this parameter, which is a typical behavior of a solid-like structure [71]. Further, because viscosity values decrease with increasing shear rate, the formulations can also be easily rubbed into the skin.

The values of the loss tangent for the formulations were 0.32, 0.32, 0.28, and 0.31, for Creams 1, 2, 3, and Control Cream, respectively, at a frequency of 0.1 Hz. All these tangent values are lower than one, which is an expected value since *G*′ > *G*″, confirming that all cream emulsions present a solid-like structure [72]. Moreover, the loss tangent values also suggest that the droplets are highly associated due to the van der Walls interactions involved in emulsion formation.

The rheological flow properties of the emulsions depend on several factors, including the droplet size and shape, their polydispersity, the interactions between the dispersed particles, and the shear conditions. The main emulsifier of these samples is a smart combination of hydrogenated phospholipids, fatty acids, and alcohols (palmitic acid and C12-16 alcohols). The saturated state of hydrogenated phospholipids and the strong packed organization of the components of this emulsifier, create a rigid lamellar structure, inducing a gel-like behavior and bringing consistency to formulas. According to Eccleston [71,73] these lamellar phases are built by the surfactant in bilayers and separated by layers of water. The hydrocarbon chains of the bilayers can exist in a number of physical states, the most relevant emulsions being the so-called ordered or gel, and the disordered, or liquid crystalline, state. The results obtained suggest that the Control Cream has a more rigid structure when compared to the extracts-based creams. This can be explained by the fact that emulsions are prepared by mixing the molten components and then cooling them to the storage temperature. At the high temperatures of manufacture, the emulsion formed by homogenization is stabilized by an adsorbed monomolecular film at the oil droplet/water interface. During cooling, the fatty amphiphile in the oil becomes increasingly insoluble and diffuses from this phase into the aqueous micelle environment to form either spherical mixed micelles or lamellar liquid crystals that further stabilize the emulsion. However, not all samples have these structures. The droplet distribution is very different for the different creams, leading to distinct rheological parameters which is probably related with the surfactant film and the phases present. The Control Cream has a more cohesive structure, with small droplets, and is more consistent, than the extract-containing creams. These differences can only be due to the presence of the extracts which, as already explained, have different compositions [72,73].

#### 3.2.4. In Vivo Compatibility, Acceptability, and Antioxidant Efficacy

The in vivo efficacy test intended to check the compatibility and acceptability of the cream, and to assess their antioxidant efficacy after application under the normal conditions of use.

In the acceptability and compatibility study, it was observed that the application of the cream formulations did not induce any adverse reactions on the skin, in any case. Thus, the formulations showed, during the study, excellent compatibility, and acceptability for the skin.

To evaluate the true changes in the b* parameter, a relative transformation in relation with D_1_ before irradiation was performed (Figure 6).

The results show that Cream 3 more successfully prevented the discoloration of β-carotene than the other creams, with a color variation (b* parameter) significantly (*p* < 0.05) lower (6.00%) than the variations in the areas treated with Cream 1 (10.42%), Cream 2 (10.72%), Control Cream (10.42%), or the untreated area (10.60%). Thus, the application of Cream 3 effectively decreases the oxidation of β-carotene under UVA irradiation. Interestingly, this formulation contains the highest concentrations of phenolic compounds HTyr and tyrosol (Table 1). The extracts, as shown in previous studies, interfere with the structure of the formulations, which can alter the bioavailability of the antioxidant substances present in the different by-products.

These results agree with previous studies that showed the antioxidant capacity of phenolic compounds obtained from olive oil by-products, namely, of HTyr. In 2004, O’Dowd et al. [20] used a specific technique to measure the H_2_O_2_ production by intact neutrophils and study how HTyr influenced this production. The results reported that HTyr exerted its antioxidant activity by scavenging H_2_O_2._ Additionally, a study conducted by Rietjens et al. [19], showed high scavenging activities of HTyr and oleuropein toward peroxynitrite.

## 4. Conclusions

The screening in vitro tests performed in this study allowed to conclude that OIBPE present elastase inhibitory activity and antioxidant capacity. The results obtained in this work also support that the OIBPE may act as UV-protection boosters to increase the absorption of some synthetic UV filters.

On the other hand, the OIBPE tested suggest their possible use in the cosmetic and/or pharmaceutical industries as preservative boosters, either separately (e.g., extract 3) or combined due to their antimicrobial activity.

Moreover, the tested samples did not show in vitro skin and eye irritation of reconstructed epidermis with normal human keratinocytes and human corneal epithelium which was confirmed in an in vivo test (HRIPT). However, only extract 2 was considered less ecotoxic. The results had shown that all extracts presented promising in vitro activities regarding the methods studied.

The olive oil by-products extracts had impact in topical creams regarding microstructure and droplet size distribution. The control cream has a more cohesive structure, with small droplets, and is more consistent than the extract-containing creams Beyond this all formulations exhibited acidic pH values suitable for topical application and shear-thinning behavior with yield stress. Additionally, the oscillatory results shown that, in all cases, the storage modulus overcome the loss modulus.

Furthermore, the in vivo efficacy tests showed excellent formulations’ compatibility and acceptability for the skin and cream 3 successfully prevented the discoloration of β-carotene being considered an antioxidant.

In conclusion, the results suggest that olive oil industry by-products extracts have valuable health properties that favor their re-use in the topical products with the advantage of overcoming environmental issues related to olive oil industry.

## Figures and Tables

**Figure 1 pharmaceutics-13-00465-f001:**
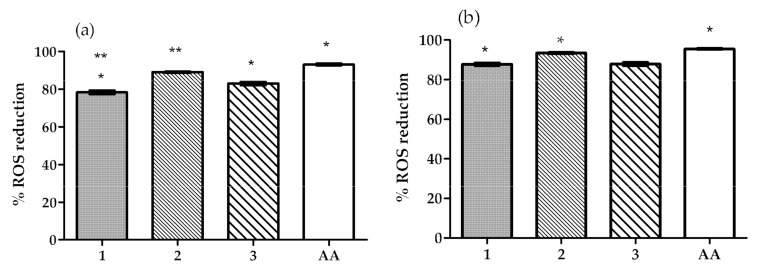
Effect of the OIBPE on the reduction of reactive oxygen species (ROS) formation (%) in HaCaT cells exposed to (**a**) hydrogen peroxide, or (**b**) UVB light for 15 min. An ascorbic acid (AA) solution (1 mg/mL) was used as positive control. Results are mean ± SD, *n* = 9 (*; ** *p* < 0.05).

**Figure 2 pharmaceutics-13-00465-f002:**
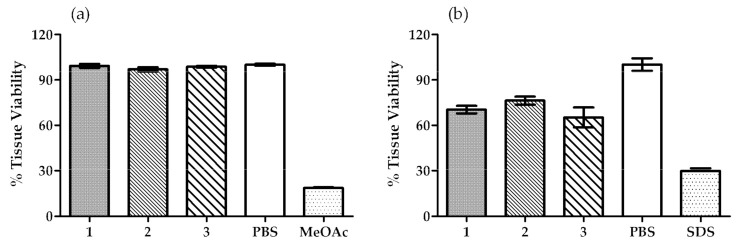
Tissue viability results obtained in (**a**) SkinEthic™—Eye irritation test, and (**b**) EpiSkin™—Skin irritation test. PBS was used as a negative control for both assays, while methyl acetate (MeOAc) and SDS were used as positive controls for SkinEthic™ and EpiSkin™, respectively. Results are mean ± SD, *n* = 2 (**a**), *n* = 3 (**b**).

**Figure 3 pharmaceutics-13-00465-f003:**
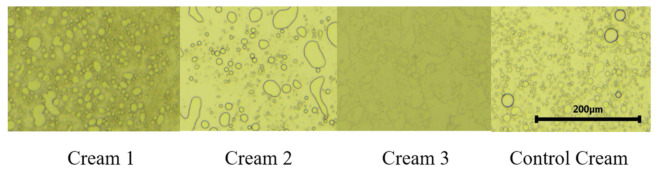
Photomicrographs of creams. Magnification: 40×.

**Figure 4 pharmaceutics-13-00465-f004:**
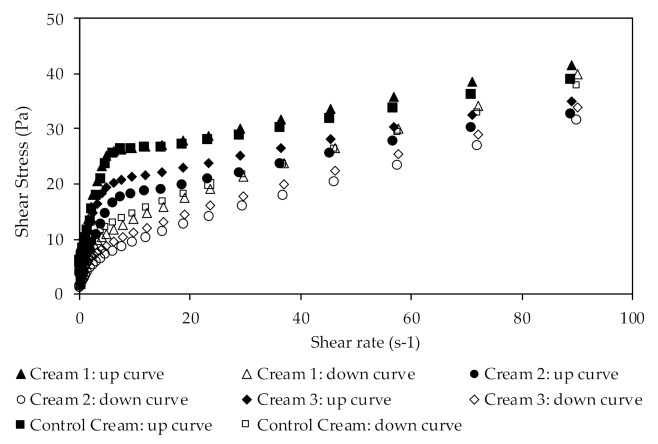
Flow curves obtained in the viscosity ramp up/ramp down assays from the extract-containing creams.

**Figure 5 pharmaceutics-13-00465-f005:**
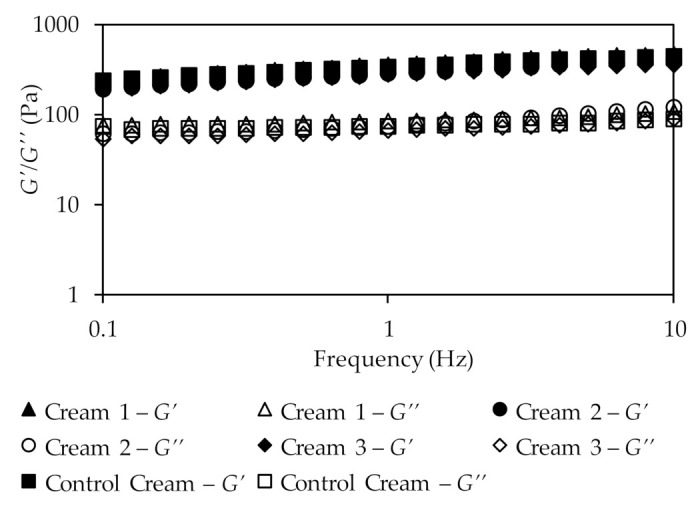
Results of the oscillatory frequency sweep test for the different creams.

**Figure 6 pharmaceutics-13-00465-f006:**
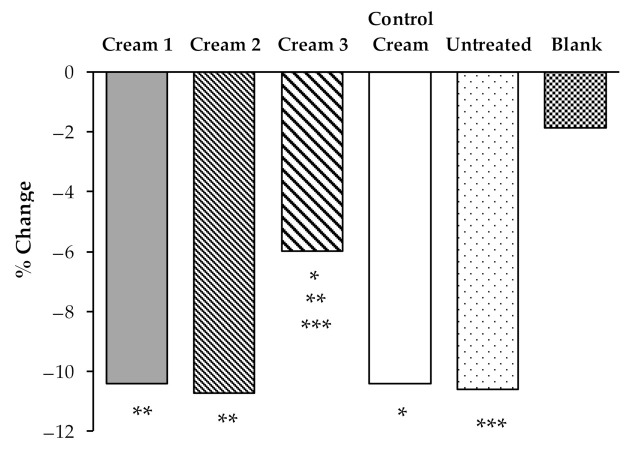
Effects of the application of the different creams on b*parameter changes before and after UVA irradiation. Mean values of all volunteers (*n* = 10). Untreated, was used as a control and a blank skin area was also used as a negative control. The statistical comparison between products and controls is also shown (*, **, ***: *p* < 0.05).

**Table 1 pharmaceutics-13-00465-t001:** Composition of the olive oil industry by-products extracts (OIBPE).

OIBPE	Composition			
	Total Phenol Compounds (mg GAE/L)	Hydroxytyrosol (mg/L)	Tyrosol (mg/L)	Oleuropein (mg/L)
1	5824	1117	134	--
2	5825	47	133	2815
3	6197	1352	113	--

**Table 2 pharmaceutics-13-00465-t002:** Qualitative and quantitative compositions of the oil-in-water (O/W) creams.

Disperse Phase	Ingredients (INCI)	Formulations (%)			
		1	2	3	Control Cream
Aqueous phase	Aqua	68.05	68.05	68.05	73.05
	Hydrogenated Lecithin (and) C12–16 Alcohols (and) Palmitic Acid	6.00	6.00	6.00	6.00
	Glycerin	5.00	5.00	5.00	5.00
	Phenoxyethanol	0.95	0.95	0.95	0.95
	OIBPE	5.00 (Extract 1)	5.00 (Extract 2)	5.00 (Extract 3)	--
Oily Phase	Caprylic/Capric Triglyceride	5.00	5.00	5.00	5.00
	Undecane (and) Tridecane	5.00	5.00	5.00	5.00
	Isopropyl Myristate	5.00	5.00	5.00	5.00

**Table 3 pharmaceutics-13-00465-t003:** IC_50_* values of olive oil industry by-products extracts in two cell lines.

OIBPE	IC_50_ (mg/mL)	
	HaCaT	L929
1	0.031 ± 0.003	0.016 ± 0.002
2	>0.1	>0.1
3	0.038 ± 0.001	0.042 ± 0.01

*IC_50_ values determined by the MTT method after 24 h of olive oil industry by-products extracts (OIBPE) incubation. Each IC_50_ value is the mean ± SD of, at least, three independent experiments with three replicates each.

**Table 4 pharmaceutics-13-00465-t004:** Enzymatic inhibition assays of elastase, collagenase, hyaluronidase, and tyrosinase by olive by-products extracts.

OIBPE	Enzymatic inhibition (%)			
	Elastase	Collagenase ^1^	Hyaluronidase	Tyrosinase ^1^
1	94 ± 2	NI	NI	10 ± 4
2	97 ± 0.3	19 ± 7	(3.1 ± 1.0) × 10	(1.3 ± 1.5) × 10
3	100 ± 0.1	9 ± 4	(2.0 ± 1.6) × 10	16 ± 6

^1^ The sample concentration used for tyrosinase was 0.0125 mg/mL and for collagenase was 0.0023 mg/mL, to overcome samples’ absorption effects at the same wavelength of the assay. (mean ± SD, n = 3) (NI—no inhibition).

**Table 5 pharmaceutics-13-00465-t005:** MIC results represented by dilutions of the extracts.

OIBPE	*S. aureus*	*B. subtilis*	*P. aeruginosa*	*C. albicans*	*E. coli*
1	N/A	1:4	1:2	N/A	1:2
2	1:4	N/A	1:8	N/A	N/A
3	1:4	1:8	1:4	1:2	1:4

N/A—No inhibition activity.

**Table 6 pharmaceutics-13-00465-t006:** EC_50_ results obtained in the ecotoxicological evaluation of the extracts.

OIBPE	Exposure Time (min)	EC_50_ (%)	95% Confidence Interval
1 (concentration: 1 × 10^−6^)	5	27.35	20.79–35.98
	15	28.07	20.47–38.48
	25	28.40	19.99–40.36
2 (concentration: 1 × 10^−3^)	5	25.15	24.50–27.91
	15	35.27	31.5–39.48
	25	35.27	32.6–38.04
3 (concentration: 1 × 10^−6^)	5	39.75	33.70–46.90
	15	41.53	33.70–51.17
	25	43.73	36.07–53.02

**Table 7 pharmaceutics-13-00465-t007:** Droplet size distribution. Results are mean ± SD, *n* = 3.

Formulation	Span	*d* (0.1)	*d* (0.5)	*d* (0.9)
Cream 1	2.66 ± 0.25	2.79 ± 0.12	34.98 ± 4.61	95.06 ± 4.49
Cream 2	1.10 ± 0.08	23.34 ± 0.97	44.34 ± 0.62	72.37 ± 3.30
Cream 3	1.74 ± 0.03	5.47 ± 0.14	43.55 ± 0.38	81.10 ± 0.56
Control Cream	3.78 ± 0.01	2.78 ± 0.10	16.33 ± 0.95	64.46 ± 3.71

**Table 8 pharmaceutics-13-00465-t008:** Fitting of mathematical models to rheological data. The Power Law and Herschel–Bulkley models were fitted to Table 2.

Formulation	Model	Yield Stress (Pa)	*K*	Chi Square	*R* ^2^
Cream 1	Power law	-	9.77, *n* = 0.313	9569	0.975
	Herschel–Bulkley	0.7097	5.48, *n* = 0.410	21.9	0.996
Cream 2	Power law	-	11.09, *n* = 0.284	10,280	0.975
	Herschel–Bulkley	1.398	4.05, *n* = 0.487	25.24	0.996
Cream 3	Power law	-	5.17, *n* = 0.371	3640	0.991
	Herschel–Bulkley	0.4718	2.41, *n* = 0.559	6.97	0.998
Control Cream	Power law	-	6.94, *n* = 0.318	4663	0.989
	Herschel–Bulkley	1.459	2.96, *n* = 0.520	13.18	0.997

## Data Availability

Not applicable.

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
