# Peer review of "Investigations of Olive Oil Industry By-Products Extracts with Potential Skin Benefits in Topical Formulations"

_pharmaceutics, 2021, doi:10.3390/pharmaceutics13040465_

Round 1
Reviewer 1 Report
Authors present an interesting investigation related to the application olive oil by-products in cosmetic formulations (creams).This is a well-known topic in the cosmetic compart, especially today because of the success of terms such as “recycle”, circular economy, by-products and renewable sources.
However the term “by -product” for olive leaves can be questionable. I can understand “product waste”, but by product “an incidental or secondary product made in the manufacture or synthesis of something else”. And this is not the case
Moreover Olive Leaf extracts are already in the market, sold by several cosmetic companies.
Maybe the extract obtained by the authors is different, but its preparation is missing and no reference about it is reported
Cytotoxicity: the results show IC50 values that are not considered during product formulation.
IC 50 values from 0.016mg/ml to 0.038 mg/ml = 0.016-0.038 g/L and the extracts are used at a concentration of 50g/L!!!
As I can understand their cytotoxicity is higher than their enzymatic activity…
As regards antimicrobical studies, the good results shown by the extracts have to be tested on the final products, using the conventional cosmetic challenge test (28 days).
As regards photoprotection, in vitro SPF is appliable to only to the final formulations. Being the extracts potential ingredients, authors had to measure their SUNSCREEN INDEX (KUMLER) AND NOT THE SHULTZ ONEs (The vehicle used strongly affect the results)
In regards of stability:
- thermal stability of the extract is missing. It has to be verified and reported because without it all the extract properties can be lost during emulsion preparation AT 80°!),
- Before the rheological studies, a study on thermal stability of the product is needed. (conventional cosmetic protocols : at least 30 days at 40°C)
In conclusion the paper must be improved from a more competent and correct cosmetic point of view.
Author Response
REVIEWER 1
Authors present an interesting investigation related to the application olive oil by-products in cosmetic formulations (creams).This is a well-known topic in the cosmetic compart, especially today because of the success of terms such as “recycle”, circular economy, by-products and renewable sources.
1-However the term “by -product” for olive leaves can be questionable. I can understand “product waste”, but by product “an incidental or secondary product made in the manufacture or synthesis of something else”. And this is not the case.
Answer: The olive oil production also includes olive crops. The olive oil production uses olives that comes with leaves and other crops. According to the Directive 200/98/CE of the European Parliament and of the Council of 19 November 2008 on waste and repealing certain Directives, that defines the conditions under which an object is to be considered a by-product. In this study the olive leaves are considered by-products.
Furthermore, a substance is considered not a waste, but a by-product when the following conditions are met:
(a) further use of the substance or object is certain – our leaves are reintegrated in our process by their deposition on soil to increase its content in organic matter and return nutrients.
(b) the substance or object can be used directly without any further processing other than normal industrial practice - applicable
(c) the substance or object is produced as an integral part of a production process - our way of olive harvesting does not allow us to harvest only olives, but at this stage we have a mixture of them with branches and leaves. The separation is performed at the mill entrance.
(d) further use is lawful, i.e. the substance or object fulfils all relevant product, environmental and health protection requirements for the specific use and will not lead to overall adverse environmental or human health impacts – applicable.
2- Moreover Olive Leaf extracts are already in the market, sold by several cosmetic companies.
Maybe the extract obtained by the authors is different, but its preparation is missing and no reference about it is reported.
Answer: The reviewer has a point. In fact, there are several olive leaf extracts in the market. However, the extracts provided by our supplier (Sovena) were produced through an innovative process and a different publication regarding the applied conditions, is being prepared.
The objective of this paper is to highlight the activity, efficacy and safety of these extracts and the impact in creams developed with them.
3 - Cytotoxicity: the results show IC50 values that are not considered during product formulation.
IC 50 values from 0.016mg/ml to 0.038 mg/ml = 0.016-0.038 g/L and the extracts are used at a concentration of 50g/L!!!
As I can understand their cytotoxicity is higher than their enzymatic activity…
Answer: The reviewer has a point. In fact, formulations were prepared at a very high concentration compared with the IC50 in monolayer cell culture of keratinocytes (HaCaT) and fibroblasts (L929), but it must be highlighted that the exposition time used in monolayer cultures was 24h, and the cells were 2D culture that does not mimic the in vivo tissue. Thus, the extracts were tested in an in vitro assay of reconstructed epidermis with normal human keratinocytes a 3D model (Epsikin model) as it is described in section section 2.2.1.2.1 and was found safe presenting a percentage of tissue viability higher than 65% when tested at 100%.
The assays in the monolayer cell lines were used during the screening assays of the extracts (results not shown).
The enzymatic inhibitory activity of several enzymes was determined at concentrations for where the cell viability was higher than 75% in in monolayer cell culture of keratinocytes (HaCaT) and fibroblasts (L929).
4- As regards antimicrobical studies, the good results shown by the extracts have to be tested on the final products, using the conventional cosmetic challenge test (28 days).
Answer:The antimicrobial studies were performed to evaluate the minimum inhibitory concentration of three extracts against for five ATCC strains, namely Staphylococcus aureus ATCC 6538, Pseudomonas aeruginosa ATCC 9027, Bacillus subtilis ATCC 6633, Escherichia coli ATCC 8739 and Candida albicans ATCC 10231.The Challenge test is applied to understand the preservative activity against microorganisms tested, which was not the main goal of this study.
5 - As regards photoprotection, in vitro SPF is appliable to only to the final formulations. Being the extracts potential ingredients, authors had to measure their SUNSCREEN INDEX (KUMLER) AND NOT THE SHULTZ ONEs (The vehicle used strongly affect the results).
Answer: The reviewer has a point, in fact SPF are usually applied to final formulations. Nevertheless, this methodology was performed as a screening test. The SPF values were measured for the OIBPE extracts using an in vitro method described by Mansur et al. [1] in which the absorption characteristics of the sunscreens agents were determined based on spectrophotometric analysis of dilute solutions. Moreover, the method uses the erythema effect spectrum (“erythema formation”) and solar intensity spectrum constant values from 290nm to 320nm, being possible to correctly accurate de in vitro SPF value of the extracts. The methods and results are described in line 203–216 and line 483-501, respectively.
[1] Mansur J de S, Breder M, Mansur MC, Azulay R. Correlação entre a determinação do fator de proteção solar em seres humanos e por espectrofotometria. An Bras Dermatol 1986;61:121–4.
The results obtained support the use of these extracts as UV-protection boosters. In this case, it is not worth doing to perform the SPF determination on final formulations.
6 - In regards of stability:
- thermal stability of the extract is missing. It has to be verified and reported because without it all the extract properties can be lost during emulsion preparation AT 80°!),
- Before the rheological studies, a study on thermal stability of the product is needed. (conventional cosmetic protocols : at least 30 days at 40°C)
Answer: The reviewer has a point. Stability studies should be performed in all formulations before market access. In the study, these formulations were used as proof of concept. The authors use an emulsifier capable of creating a rigid lamellar structure, inducing a gel-like behavior and bringing consistency to formulas. The results obtained revealed no crystalline structures meaning that lamellar structures were not present. Said that, it may be hypothesized that these results may be due to interactions between components of the incorporated extracts and the formulation, which might change their internal structure and, consequently, their stability.
In conclusion the paper must be improved from a more competent and correct cosmetic point of view.
Reviewer 2 Report
Dear Authors,
I personally enjoyed the reading of the manuscript. I would like to highlights some minor revisions that in my opinion needs to be implemented before proceeding toward publication.
Lines 203-215
The photoprotection was inquired in a quite small portion of the spectrum. Indeed the lower UV region (UVC) was not tested. The authors should make some remarks on this since it is the characteristic wavelength which is more harmful for the human skin.
Also it is not clear which blank was used for this test
Table 3 and 4
Make sure that there spacing provided between the table and description ( 1 line blank)
Lines 474-477
The authors pointed out that the proposed approach for antioxidant activity calculation was different respect to more classical ones. The authors should better present the rationality of using the proposed novel methodologies. Also in describing the standard antioxidant the authors mentioned ABTS and DPPH, I suggest them to complete this methodological review by mentioning also the use of EPR spin trapping for the evaluation of skin formulations.
Lines 482-489
I suggest the authors to plot the data reported with an histogramm pointing out the difference in the phenolic concentration which was reported to cause the differences in the photoprotection properties.
Lines 574-579
This paragraph is not in my opinion necessary since it just repeat the experiment done but does not report any result.
lastly please check the spelling and typos one more time I found a few in the manuscript like florescence in place of fluorescence
Author Response
REVIEWER 2
I personally enjoyed the reading of the manuscript. I would like to highlights some minor revisions that in my opinion needs to be implemented before proceeding toward publication.
1- Lines 203-215 The photoprotection was inquired in a quite small portion of the spectrum. Indeed the lower UV region (UVC) was not tested. The authors should make some remarks on this since it is the characteristic wavelength which is more harmful for the human skin.
Answer: The UVA radiation is responsible for the premature aging of the skin and can act as a co-carcinogen with UVB radiation, since it reaches the deeper layers of the epidermis and dermis. UVB radiation is not completely filtered out by the ozone layer and is responsible for sunburn and can induce skin cancer. UVC radiation is highly carcinogenic, but is mostly absorbed by the ozone-rich stratosphere, being filtered by the atmosphere before reaching earth. [1-3] Therefore, we used a method described by Mansur et al. [4] in which the absorption characteristics of the sunscreens agents are determined based on spectrophotometric analysis of dilute solutions. Moreover, the method uses the erythema effect spectrum (“erythema formation”) and solar intensity spectrum constant values, being possible to correctly accurate de in vitro SPF value of the extracts.
[1] Marto J, Gouveia L, Chiari B, Paiva A, Isaac V, Pinto P, et al. The green generation of sunscreens: Using coffee industrial sub-products. Ind Crops Prod 2016;80:93–100. https://doi.org/10.1017/CBO9781107415324.004.
[2] Dutra EA, Da Costa E Oliveira DAG, Kedor-Hackmann ERM, Miritello Santoro MIR. Determination of sun protection factor (SPF) of sunscreens by ultraviolet spectrophotometry. Rev Bras Ciencias Farm J Pharm Sci 2004;40:381–5. https://doi.org/10.1590/S1516-93322004000300014.
[3] Venus M, Waterman J, McNab I. Basic physiology of the skin. Surgery 2011;29:471–4. https://doi.org/10.1016/j.mpsur.2011.06.010.
[4] Mansur J de S, Breder M, Mansur MC, Azulay R. Correlação entre a determinação do fator de proteção solar em seres humanos e por espectrofotometria. An Bras Dermatol 1986;61:121–4.
2 - Also it is not clear which blank was used for this test
Answer: The reviewer has a point. Distilled water was used as control. Text changed as indicated.
3 -Table 3 and 4
Make sure that there spacing provided between the table and description ( 1 line blank)
Answer: Text changed as indicated.
4 - Lines 474-477 The authors pointed out that the proposed approach for antioxidant activity calculation was different respect to more classical ones. The authors should better present the rationality of using the proposed novel methodologies. Also in describing the standard antioxidant the authors mentioned ABTS and DPPH, I suggest them to complete this methodological review by mentioning also the use of EPR spin trapping for the evaluation of skin formulations.
Answer: The full chemical and properties characterization of these extracts is not yet published as explained in lines 92-94: “OIBPE were a gift from Sovena Group. In previous studies, the extracts exhibited health-promoting properties, possibly due to their antioxidant, anti-inflammatory, photoprotector and antimicrobial activities (data not shown).” Actually, different methodologies were used to confirm the antioxidant activity, namely, DDPH, ABTS and probably EPR spin trapping. In this manuscript the authors decided to publish data obtained from in vitro antioxidant assay, using cells.
5 - Lines 482-489 I suggest the authors to plot the data reported with an histogram pointing out the difference in the phenolic concentration which was reported to cause the differences in the photoprotection properties.
Answer: As mentioned in the article, no relationship could be observed between the SPF of the extracts and their concentrations of phenolic compounds. Although, the SPF values may be related to the minor compounds found in the extracts. Due to the infinity of this compounds and their analysis complexity, there wasn’t any possibility to identify each minor compound and correlate all data.
6 -Lines 574-579 This paragraph is not in my opinion necessary since it just repeat the experiment done but does not report any result.
Answer: The reviewer has a point. Nevertheless, this paragraph was included to contextualize all the results regarding Formulations containing olive oil industry by-products extracts
7 - lastly please check the spelling and typos one more time I found a few in the manuscript like florescence in place of fluorescence
Answer: Text changed as indicated.
Reviewer 3 Report
Dear authors,
the submitted manuscript represents a comprehensive evaluation of olive oil industry by-products utilization potential. The study was well designed and conducted, thus resulting in valuable conclusions. As of my concern, the only issue represents not including positive controls in cytotoxicity assays. Therefore, I would suggest, if it is possible, to perform the described assay with some certified cytotoxic drug (not SDS at conc. 1mg/mL, which has GRAS status)
Kind regards
Author Response
REVIEWER 3
the submitted manuscript represents a comprehensive evaluation of olive oil industry by-products utilization potential. The study was well designed and conducted, thus resulting in valuable conclusions.
As of my concern, the only issue represents not including positive controls in cytotoxicity assays. Therefore, I would suggest, if it is possible, to perform the described assay with some certified cytotoxic drug (not SDS at conc. 1mg/mL, which has GRAS status)
Answer: The reviewer has a point. In fact the use of a certified cytotoxic drug structural close to the ones we are testing would be the ideal to be used as positive control in cytotoxicity assays. In this work the cytotoxicity assay was done using the same negative and positive controls as those used in the 3D model (described in section 2.2.1.2.1.) a validated method that is based on the Reconstructed Human Epidermis (RHE) model, which was granted regulatory approval as a full replacement for the rabbit in vivo skin irritation test under the EC Test Method Regulation (Method B.46, EU, 2009) and under conditions laid down in OECD Test Guideline 439 (OECD, 2010). The test consists of a topical exposure of the neat test chemical to a reconstructed human epidermis (RHE) model, followed by a cell viability test.
Round 2
Reviewer 1 Report
I suggest to somehow insert in the text their replies to my comments as explanations regarding the choice of the experiments presented (the choice of the in vitro spf, antimicrobial activity, the choice of high concentrations of the extract in the formulations etc), the reason why the extract composition and extractive method is not reported and the advantages of this kind of extract when compared to commercial ones
I still keeping that the work has not been carried out from a useful cosmetic point of view and without changes it cannot be accepted
Author Response
Dear Reviewer,
Thank you for the suggestions and the opportunity we have been given to improve the quality of our work. We proceeded to revise the manuscript according to the points raised by you and hope the corrections meet your expectations.
Answers
Text changed as indicated and highlighted in yellow.
1 - The OIBPE were a gift from Sovena Group. In previous studies, the extracts exhibited health-promoting properties, possibly due to their antioxidant, anti-inflammatory, photoprotector and antimicrobial activities (data not shown). Since the olive harvesting includes a mixture of olives with branches and leaves that are used directly without any further processing and leaves are reintegrated in olive oil process by their deposition on soil to increase its content in organic matter and return nutrients there they are considered by-products. The OIBPE were obtained from leaves, without using organic solvents, either at high temperature (extracts 1 and 3) or at room temperature (extract 2). The extracts provided by our supplier (Sovena) were produced through an innovative process that are not shown due to confidentiality reasons.
2 - To evaluate the photoprotector activity of the OIBPE, a spectrophotometry method was used to calculate the sun protection factor (SPF), as described by Mansur et al. [50], as a screening test. This methodology measures the absorption characteristics of the sunscreens agents based on spectrophotometric analysis of dilute solutions. Moreover, the method uses the erythema effect spectrum (“erythema formation”) and solar intensity spectrum constant values from 290nm to 320nm, being possible to correctly accurate de in vitro SPF value of the extracts.
The OIBPE were diluted (1:40) in distilled water in a 96-well plate (n = 3). The absorption spectrum was measured from 290 nm to 320 nm every 5 nm, with 3 measurements per point. Distilled water was used as control. The spectra were obtained using a fluorescence microplate reader (FLUOstar BMGLabtech, Germany.
3 -The antimicrobial studies were performed to evaluate the minimum inhibitory concentration of three extracts against for five ATCC strains, namely Staphylococcus aureus ATCC 6538, Pseudomonas aeruginosa ATCC 9027, Bacillus subtilis ATCC 6633, Escherichia coli ATCC 8739 and Candida albicans ATCC 10231.
4 - Three O/W creams were prepared using three different OIBPE, as well as a control formulation consisting of a cream without any added extract. The 5% OIBPE concentration was used as the worst possible scenario regarding the impact of these extracts on the structure of the formulations.
5 - In conclusion, the results suggest that olive oil industry by-products extracts have valuable skin health properties that favor their re-use in the topical products with the advantage of overcoming environmental issues related to olive oil industry.
Round 3
Reviewer 1 Report
Ok.